# Oxidative Damages on the Alzheimer’s Related-Aβ Peptide Alters Its Ability to Assemble

**DOI:** 10.3390/antiox12020472

**Published:** 2023-02-13

**Authors:** Clémence Cheignon, Fabrice Collin, Laurent Sabater, Christelle Hureau

**Affiliations:** LCC-CNRS, Université de Toulouse, CNRS, 31077 Toulouse, France

**Keywords:** oxidative stress, peptide, assembly, copper, zinc

## Abstract

Oxidative stress that can lead to oxidation of the amyloid-β (Aβ) peptide is considered a key feature in Alzheimer’s disease (AD), influencing the ability of Aβ to assemble into β-sheet rich fibrils that are commonly found in senile plaques of AD patients. The present study aims at investigating the fallouts of Aβ oxidation on the assembly properties of the Aβ peptide. To accomplish this, we performed kinetics and analysis on an oxidized Aβ (^ox^Aβ) peptide, resulting from the attack of reactive oxygen species (ROS) that are formed by the biologically relevant Cu/Aβ/dioxygen/ascorbate system. ^ox^Aβ was still able to assemble but displayed ill-defined and small oligomeric assemblies compared to the long and thick β-sheet rich fibrils from the non-oxidized counterpart. In addition, ^ox^Aβ does affect the assembly of the parent Aβ peptide. In a mixture of the two peptides, ^ox^Aβ has a mainly kinetic effect on the assembly of the Aβ peptide and was able to slow down the formation of Aβ fibril in a wide pH range [6.0–7.4]. However, ^ox^Aβ does not change the quantity and morphology of the Aβ fibrils formed to a significant extent. In the presence of copper or zinc di-cations, ^ox^Aβ assembled into weakly-structured aggregates rather than short, untangled Cu-Aβ fibrils and long untangled Zn-Aβ fibrils. The delaying effect of ^ox^Aβ on metal altered Aβ assembly was also observed. Hence, our results obtained here bring new insights regarding the tight interconnection between (i) ROS production leading to Aβ oxidation and (ii) Aβ assembly, in particular via the modulation of the Aβ assembly by ^ox^Aβ. It is the first time that co-assembly of ^ox^Aβ and Aβ under various environmental conditions (pH, metal ions …) are reported.

## 1. Introduction

Alzheimer’s disease (AD) is one of the most, if not the most, well-known amyloid-related diseases. Amyloids refer to a specific arrangement of very stable, intrinsically disordered peptides by the alignment of β-sheets perpendicular to the longer axis [1,2,3]. Given this, amyloid deposits are found in a large panel of pathologies, such as AD, Type-II diabetes Mellitus, Parkinson’s disease and systemic amyloidosis [4,5]. In AD, the senile plaques made of the extracellular deposits of Amyloid-β (Aβ) peptides in the fibrillar state (*id est*, as amyloids) is the first pathological hallmark of the disorder [6,7,8], whilst the second one is the intraneuronal neurofibrillary tangles made of hyperphosphorylated Tau protein [9,10,11].

The assembly processes leading to the formation of amyloids are extremely complex, featuring many steps and species at play [2,3,12]. A simplified view is given in Figure 1, wherein the formation of fibrils is highlighted and pathways leading to the formation of less-structured or amorphous species are also illustrated. It is worth noting that the in vitro assays on amyloid proteins are thoroughly described in the literature as mirrors of the in vivo assembly processes, and that in vitro investigations of assembly represent a powerful tool to better understand in vivo assembly and are a crucial step to understand the biological mechanisms involved [13,14]. The self-assembly occurs through nucleation/elongation/secondary nucleation paths [12,14,15,16,17,18]. Nucleation is the formation of high-energy and low-molecular weight soluble intermediates from the monomers, wherein several kinds of intermediates can co-exist. Among them, metastable partially folded oligomers and nuclei can further elongate at their extremities. On the other hand, there are secondary nucleation processes that make the self-assembly of peptides auto catalytic [14,15]. Two main pathways can co-exist. The former is independent of the monomers concentration and corresponds to fragmentation of the fibrils, leading to shorter fibrils that can further elongate; the latter consists of fibrils-catalyzed nucleation, depending on the monomers concentration, and induces the formation of new nuclei by interaction of the monomers with the surface of the fibrils [19]. Furthermore, the self-assembly of amyloid-forming peptides depends on several parameters. The main parameter is the sequence of the peptide itself, but environmental factors, such as pH, can also affect these processes [20]. In the case of the Aβ peptides, the cleavage from the Amyloid Protein Precursor (APP) by β- and γ-secretases is heterogeneous, thus generating peptides of different lengths (up to 42 amino-acids). Aβ_1–40_ and Aβ_1–42_, 40 and 42 amino-acid residues long peptides are the most studied ones in the context of AD. Both N-terminal modification and C-terminal truncation impact Aβ self-assembly ability. Beyond that, Aβ peptides of different lengths can co-assemble along several mechanisms [21,22,23,24,25,26,27,28]. Recently, we found that Aβ_4–40_ and Aβ_1–40_ co-assemble faster than either peptide self-assembles independently [21]. Other examples include the co-assembly of Aβ_1/11–40_ with Aβ_1–42_, that proceeds as two independent self-assemblies [22,23], whilst Aβ_11–40_ can recruit Aβ_1–40_ in a process where Aβ_11–40_ imposes its assembly properties [22] and Aβ_5–42_ can seed the self-assembly of Aβ_1–42_ [24]. More recently, the co-assembly of genetic Artic and Italian variants with Aβ_1–40/42_ were also studied [25]. Hence, these various co-assembly processes need to be taken into account to mirror, as much as possible, the intrinsic biological complexity.

Another key feature of AD is the post-mortem detection of oxidative damages, not only occurring on the biomolecules in the surroundings of the amyloid deposits, but also on the Aβ peptides themselves [29]. Indeed, the ability of the Aβ-bound Cu (Cu-Aβ) to generate reactive oxygen species (ROS) can contribute to the oxidative stress fallouts detected in AD patients. It is mainly due to the presence of an exchangeable pool of Cu (about 1–10 µM) [30] and ascorbate at a fairly high level (about 100–300 µM), in the synaptic cleft [31,32,33], that can fuel the incomplete reduction of dioxygen to ROS, catalyzed by Cu-Aβ [29]. The Cu-Aβ-induced oxidative modifications of Aβ_1–40_ described in the literature are illustrated in Figure 2. Amino acid residues involved in Cu(I) and Cu(II) coordination [30] were found to be the main targets of ROS produced at the metal center: Asp1 oxidation and/or oxidative cleavage [34,35,36,37] and His13/His14 oxidation to oxo-histidine [34,35,36,37,38,39,40,41]). Phe19/20 [34,35,38] and Met35 [35,37,39] oxidations were also reported. Furthermore, tyrosine oxidation generating a dityrosine unit was only detected in a few studies [42,43,44]. This is probably because it is a minor oxidation, as reported recently [44,45], and/or due to the intrinsic detection challenge [45]. Other oxidative cleavages were also previously reported [36] at Ala2/Glu3, Val12/His13 and His13/His14 positions.

In addition to the redox-active Cu ions, Zn(II) was found at 10–100 µM in the synaptic cleft. Both ions were detected in the amyloid plaques at millimolar level and were described as modulators of Aβ self-assembly in vitro, leading to Aβ assemblies of different morphologies, which led to various neurotoxic effects. Despite no strong consensus on the in vitro data, several points of convergence [30,46,47,48] support that (i) the effects of Cu(II) and Zn(II) on Aβ assembly are metal-dependent and metal-peptide ratio dependent and (ii) at the sub-molar level (versus Aβ_1–40_), Cu(II) and Zn(II) mainly impact kinetics of amyloid formation and delay the assembly process, whilst at 1:1 stoichiometric and higher ratio, Cu(II) and Zn(II) impact the morphologies of the assemblies formed. Cu(II) favors shorter and thinner amyloids, while Zn(II) fosters ill-defined and amorphous aggregates.

Additionally, iron impairment is another contributor to AD pathology, which was found in senile plaques [49,50]. However, its exact role is not fully understood [50,51] and its speciation is not clear, with nanoparticles of mainly Fe, such as magnetite or ferritin-based minerals, being detected in the core of the plaques [49]. In contrast to Cu and Zn for which the molecular interaction with Aβ was characterized, only a few studies proposed a possible coordination site for the Fe(II)-Aβ [52] or Fe(III)-Aβ complex [53]. Despite the fact that the Fe(II)/Fe(III) redox couple may participate to the oxidative stress linked to AD [54,55], there is no evidence that this may be due to the Fe(II)/Fe(III)-Aβ interaction. There is also no study on the Fe(II) impact on the Aβ self-assembly, even if rare findings in this context were obtained for Fe(III) [56,57,58,59] or ferritin nanoparticles [60]. Therefore, the present report is focused on the impact of the metal ions Cu(II) and Zn(II) on the assembly of Aβ.

A relationship between the self-assembly of Aβ_1–40_ and Cu-Aβ-induced ROS production ability was previously shown. On the one hand, it has been reported that Cu-Aβ_1–40_ intermediate-size assemblies and fibrils produce less ROS than the Cu-Aβ_1–40_ monomer, a propensity that is shared by α-synuclein, another amyloid-forming peptide involved in Parkinson’s disease [61,62]. On the other hand, the mechanisms leading to site-specific oxidation of Aβ that affect its assembly propensity were only investigated in a few studies, mainly in non-physiological conditions, such as cold atmospheric plasma-induced oxidation [63] and hydroxyl radical-based fast photochemical oxidation [64], or focused on one specific amino acid residue oxidation, such as Met35 oxidation by hydrogen peroxide [65,66,67,68] and tyrosine oxidation, leading to dimer formation [43,44,69]. Dityrosine cross-links were shown to impede fibril formation and form soluble aggregates and/or short fibrils in the presence of copper [43,44,69]. Conversely, the effect of oxidized Met35 in Aβ_1–40_ peptide is less clear. Several trends were reported: slower assembly but no effect on the morphology of the aggregates [65], shorter lag-time of fibril formation [67,68] along with the promotion of fragmented fibrils [67] and a lessening of oligomers formation [66].

In addition to those studies where oxidation at one specific amino-acid residues were explored, our main focus was to study (i) whether and to what extent the biologically relevant combination of oxidized peptides (noted ^ox^Aβ_1–40_) [31,34,35,70,71], formed by the Cu-Aβ/ascorbate/dioxygen triad [35], self-assembles and modulates the assembly of Aβ_1–40_ and (ii) the difference in the metal-modulated assembly of ^ox^Aβ_1–40_ compared to Aβ_1–40_ peptides. Given this, this work contributes to delineate the impact of physiologically relevant oxidative damages on the assembly of Aβ peptide to close the loop between Cu-Aβ ROS production and Aβ assembly, the two most important molecular and supra-molecular events related to the etiology of AD.

In the present article, we report on the self-assembly of ^ox^Aβ_1–40_, Aβ_1–40_ and a mixture of them in various ratios, at several pH, and compare Cu and Zn-modulated aggregation of Aβ_1–40_ and ^ox^Aβ_1–40_ using kinetics and imaging experiments, whilst the detailed analysis of ^ox^Aβ_1–40_ formation was previously reported [35]. Such fundamental and basic research is fully required [72] to fully understand the molecular interactions responsible for Aβ assembly, wherein the close relationship between oxidative stress and Aβ self-assembly may play a major role in AD pathology [29,73,74].

## 2. Materials and Methods

### 2.1. Chemicals

A total of 0.1 M stock solutions of Cu(II) and Zn(II) (from CuSO_4_.5(H_2_O) and ZnSO_4_(H_2_O), respectively, purchased from Sigma (St. Louis, MO, USA)) were prepared in ultrapure water. Phosphate buffer was bought from Sigma-Aldrich (St. Louis, MO, USA) and dissolved in ultrapure water to reach a 0.1 M concentration. Bioluminescence grade HEPES buffer (sodium salt of 2-[4-(2-hydroxyethyl)piperazin-1-yl]ethanesulfonic acid) was bought from Honeywell Fluka (Morristown, NJ, USA) and dissolved in ultrapure water to reach a 0.5 M concentration. Tris hydrochloride (Tris-HCl) was purchased from Fluka and dissolved in ultrapure water to reach 1 M concentration, and was adjusted with NaOH to pH 11.0. Guanidinium chloride >99% was bought from Alfa Aesar (Haverhill, MA, USA) and was freshly prepared by dissolving the powder in Tris-HCl 0.1 M to reach a 6 M concentration. A 5 mM ascorbate solution was freshly prepared a few minutes prior to each experimental set by dissolving sodium L-ascorbate (Sigma) in ultrapure water. Ethylene diamine tetraacetic acid (EDTA) was purchased from Sigma-Aldrich and dissolved in ultrapure water to reach a 40 mM concentration. A stock solution of Thioflavin T (ThT) at 250 μM was prepared in water without any further purification, with ThT bought from Acros Organics (Waltham, MA, USA).

### 2.2. Peptide Preparation

All the synthetic peptides were bought from GeneCust (Dudelange, Luxembourg), with purity grade > 95%. Stock solutions of the Aβ_1–40_ (sequence DAEFRHDSGYEVHHQKLVFFAEDVGSNKGAIIGLMVGGVV) peptide were prepared by dissolving the powder (~3 mg) in 500 µL of Tris-HCl (0.1 M) with Guanidinium chloride (6 M). The solutions were incubated at 20 °C overnight and purified by Fast Protein Liquid Chromatography (FPLC) (column Superdex 75, elution solvent NaOH 15 mM with NaCl 150 mM, flow rate 0.5 mL/min). The peptide concentration in the recovered fractions (500 µL) was then determined by UV-visible absorption of Tyr10, considered as free tyrosine (at pH 12, (ε_293_ − ε_360_) = 2400 M^−1^ cm^−1^). The solution was used as soon as possible in ThT fluorescence experiments.

### 2.3. Peptide Oxidation

A stock solution of Aβ_1–40_ peptide was made fresh by dissolving the powder in 15 mM NaOH and titrated by UV-visible absorption of Tyr10, considered as free tyrosine (at pH 12, (ε_293_ − ε_360_) = 2400 M^−1^ cm^−1^). The Aβ_40_ peptide (60 µM) was oxidized in a phosphate buffered solution (50 mM, pH 7.4) containing Cu(II) (50 µM) and ascorbate (0.5 mM) for 30 min. Then, the solution was concentrated with an Amicon Ultra 3 kDa membrane (Millipore, Burlington, MA, USA), washed with EDTA (10 equivalents) to remove copper and then washed with water. The oxidized peptide solution was recovered, incubated at 20 °C overnight with Tris-HCl (0.1 M) with Guanidinium chloride (6 M) [pH 10] and purified by Fast Protein Liquid Chromatography (FPLC) (column Superdex 75, elution solvent NaOH 15 mM with NaCl 150 mM, flow rate 0.5 mL/min). The peptide concentration in the recovered fractions (500 µL) was then determined by UV-visible absorption of Tyr10, considered as free tyrosine (at pH 12, (ε_293_ − ε_360_) = 2400 M^−1^cm^−1^). Since the oxidized peptide solution has a background absorbance at 293 nm, the curve is fitted to subtract the absorbance due to the tailing from the Tyr absorption, as previously described [70]. The solution was used as soon as possible in ThT fluorescence experiments.

### 2.4. ThT Assay for Aβ_1–40_ and ^ox^Aβ_1–40_ Aggregation

Fluorescence experiments were performed on a FLUOstar Optima microplate reader system (BMG Labtech, Ortenberg, Germany) at 37 °C. Thioflavin-T (ThT) was used as a probe for β-sheet structures’ formation [75,76]. Fluorescence was measured every 5 min during about 120 to 160 h (depending on the experiment), after 15 s of shaking at 200 rpm. A total of 384-well microplates were used, with a total volume of 100 µL for each sample. The time course of ThT fluorescence was then measured (excitation at 440 ± 10 nm, emission at 490 ± 10 nm). For experiments in the presence of metal ions, Cu(II) or Zn(II) was added in the solution as the last reagent. All the conditions were not systematically performed on the 4 independent experiments (the quantity of peptide available for one experiment is limited).

### 2.5. Transmission Electron Microscopy (TEM)

Solutions were collected from the fluorescence microplate after 120 to 150 h and prepared for TEM using the conventional negative staining procedure. An amount of 20 μL of the solution was adsorbed on Formvar-carbon-coated grids for 2 min, blotted and negatively stained with uranyl acetate (1%) for 1 min. Grids were examined with a TEM (Jeol JEM-1400, JEOL Inc, Peabody, MA, USA) at 80 kV. Images were acquired using a digital camera (Gatan Orius, Gatan Inc, Pleasanton, CA, USA) at 10,000× or 25,000× magnification. Since the phosphate buffer reacts with uranyl acetate, HEPES buffer was preferred and employed for TEM experiments, and thus, for ThT fluorescence experiments as well.

### 2.6. Determination of the Percentage of Remaining Non-Oxidized Peptides in the Solution of ^ox^Aβ_1–40_ Peptide

The remaining Aβ_1–40_ was evaluated by HPLC/HR-MS (Dionex Ultimate 300O coupled to LTQ-Orbitrap, ThermoScientific, Waltham, MA, USA). A total of 5 µL of the control Aβ_1–40_ (non-oxidized) and ^ox^Aβ_1–40_ were injected onto the column (Acclaim 120 C18, 50 × 3 mm, 3 µm, ThermoScientific) at room temperature. The gradient elution was carried out with formic acid 0.1% (mobile phase A) and acetonitrile/water (80/20 *v*/*v*) formic acid 0.1% (mobile phase B), at a flow rate of 500 µL min^−1^. The mobile phase gradient was programmed with the following time course: 5% mobile phase B at 0 min, held 3 min, linear increase to 55% B at 8 min, linear increase to 100% of B at 9 min, held 2 min, linear decrease to 5% B at 12 min and held 3 min. The mass spectrometer was used as a detector, working in the full scan positive mode between m/z 150 and 2000, at a resolution power of 60,000. Extracted chromatograms (accuracy 5 ppm) were obtained for the most intense ions of Aβ_1–40_ in our experimental conditions, i.e., [M+4H]^4+^ and [M+5H]^5+^ (respectively detected at m/z 1082.7949 and m/z 866.4375). According to ref [35], chromatographic peaks were integrated, and the remaining Aβ_1–40_ in ^ox^Aβ_1–40_ samples was found to be in the 10–40% range, depending on the performed experiment, as illustrated in Appendix A for experiment 1, for which 15 ± 3% of the remaining Aβ_1–40_ was evaluated.

## 3. Results

### 3.1. Oxidative Damages Leading to ^ox^Aβ_1–40_

The detailed and full kinetic analysis of the oxidative damages undergone by the Aβ_1–40_ peptide have been previously reported [35]. Hence, the characterization of the oxidation of the Aβ_1–40_ peptide is given in the Appendix A. The results are fully consistent with those previously reported on another peptide batch [35], which is in line with the correct replicability of such a Cu-Aβ/ascorbate/dioxygen oxidation procedure [34].

### 3.2. Self-Assembly of Aβ_1–40_ and ^ox^Aβ_1–40_ and Their Co-Assembly

Figure 1 shows the fluorescence enhancement of the Thioflavin-T (ThT) dye upon assembly of Aβ_1–40_ and ^ox^Aβ_1–40_, and of a 1:1 stoichiometric mixture of them, for four independent experiments. ThT fluorescence is currently the golden standard to evaluate the formation of β-sheet rich supramolecular architectures, due to the strong enhancement of its fluorescence (by about a 10^4^-fold) upon interaction with amyloids [77,78]. This is fully appropriate for the screening of various conditions on plate fluorimeters. Of important note, it is well described that having reproducible assembly data for amyloid-forming peptides, such as Aβ, is highly challenging [12,79], but this is a pre-requisite to efficiently discuss assembly trends, as well as the effects of the aggregation’s modifiers. Given this, we report four independent studies (termed experiments N°1 to 4) on which the conditions were performed, at least in triplicates. For Aβ_1–40_ and ^ox^Aβ_1–40_, ThT fluorescent kinetic traces show the classical s-shape curves corresponding to the nucleation-elongation supramolecular polymerization process, as previously described (Figure 1). Several mathematical models have been proposed to reproduce such curves, the simplest one being given in Equation (1), while others that are more sophisticated have been reported to consider the asymmetry of the curve or double-sigmoidal process [12,21,80,81].
(1)$$F\left(t\right)=F_{0}+\frac{F_{max}-F_{0}}{\left(1+exp^{-k\left(t-t_{1/2}\right)}\right)}\;\mathrm{and}\;t_{lag}=t_{1/2}-\frac{2}{k}$$
where $F_{0}$ is the starting ThT fluorescence value, $F_{max}$ is the maximum of ThT intensity, $t_{1/2}$ is the time at which the ThT fluorescence equals $\frac{F_{max}+F_{0}}{2}$ and k is the elongation rate.

We observed two main features (Figure 1): (i) Aβ_1–40_ (black lines) and ^ox^Aβ_1–40_ (blue lines) show very different self-assembly trends, with Aβ_1–40_ having a decreased t_1/2_ (2 to 10-fold) and much higher F_max_ values (2.5 to 4-fold) than ^ox^Aβ_1–40_, consistently observed in the four independent experiments reported. However, the quantitative differences between Aβ_1–40_ and ^ox^Aβ_1–40_ vary from one experiment to another, hypothesized to be due to the level of remaining non-oxidized Aβ_1–40_ in ^ox^Aβ_1–40_ samples (about 10% in experiments 1 and 3 and about 30 and 40% in experiments 4 and 2, respectively; see Material and Methods for details); (ii) the comparison of assembly from a 1:1 stoichiometric mixture of Aβ_1–40_ and ^ox^Aβ_1–40_ (10 µM each, green lines), or from the control with Aβ_1–40_ only (10 µM, red lines), shows that ^ox^Aβ_1–40_ induces a slowdown of the Aβ_1–40_ assembly that cannot be driven by dilution effect only. The extent of this effect is even more obvious in experiments 1 and 3 when compared to experiments 2 and 4. For the latter ones, the levels of Aβ_1–40_ and ^ox^Aβ_1–40_ are under- and over-estimated, respectively, since some non-oxidized Aβ_1–40_ is present in ^ox^Aβ_1–40_ samples. The higher real [Aβ_1–40_] and lower real [^ox^Aβ_1–40_] (up to 13–15 µM and down to 5–7 µM in these two experiments) counter-balanced the slow down effect of ^ox^Aβ_1–40_. In addition, the F_max_ values for both conditions ([Aβ_1–40_] = 10 µM + [^ox^Aβ_1–40_] = 10 µM and [Aβ_1–40_] = 10 µM) are similar, suggesting that, despite the ^ox^Aβ_1–40_ having a kinetic effect on Aβ_1–40_ assembly, it is not recruited to form fibrils together with Aβ_1–40_, which would have led to an increase in ThT fluorescence intensity.

Beyond kinetic differences in their formation mechanisms, the final species obtained from the assembly of Aβ_1–40_ and ^ox^Aβ_1–40_ show different morphologies. TEM pictures taken at the end of Aβ_1–40_ assembly display long, mature fibrils ranging from 200 nm to 1 µm in length (Figure 2A). For ^ox^Aβ_1–40_, several kinds of aggregates with different morphologies are observed, including mainly amorphous species, but also oligomers and shorter and thinner fibrils (Figure 2B). This is in line with the lower final fluorescence intensity of ^ox^Aβ_1–40_ compared to Aβ_1–40_. It is also worth noting that fibrils may come from the remaining unoxidized peptide in the ^ox^Aβ_1–40_ sample. These results suggest that ^ox^Aβ_1–40_ keeps the ability to self-assemble, but with lower propensity to form fibrils than Aβ_1–40_, which is in line with the weaker ThT intensity. In the presence of both ^ox^Aβ_1–40_ and Aβ_1–40_, amorphous aggregates are found together with fibrils longer and thinner than those observed in the absence of ^ox^Aβ_1–40_ (Figure 2D). This is not only a dilution effect since no ill-defined assemblies are observed when [Aβ_1–40_] = 10 µM (Figure 2C). Overall, TEM pictures indicate that ^ox^Aβ_1–40_ and Aβ_1–40_ may form independent assemblies, which is in line with the ThT kinetic data.

### 3.3. Impact of pH on the Self-Assembly of Aβ_1–40_ and ^ox^Aβ_1–40_ and Their Co-Assembly

The effect of pH on the assembly of Aβ_1–40_, ^ox^Aβ_1–40_ and of a mixture at 1:1 stoichiometric ratio was investigated in experiment N°1 (Figure 3) and experiment N°3 (Appendix A). We found different pH-dependent effects on Aβ_1–40_ and ^ox^Aβ_1–40_ self-assembly. From Aβ_1–40_ kinetics assessment, pH mainly modifies the shape of the s-curve, while t_1/2_ and F_max_ values are almost unaffected (less than 20%). With respect to the shape of the curve, this result is in good agreement with a recent seminal and thorough study focusing on the origin of the pH-dependent assembly of Aβ_1–40_ [20]. Concerning the t_1/2_ value, our results are in contrast with previously published work [20], which may be attributed to the four times higher concentration we used. From ^ox^Aβ_1–40_ kinetics assessment and, conversely, to Aβ_1–40_, the t_1/2_ values of the ^ox^Aβ_1–40_ self-assembly strongly depends on the pH, with an increase of t_1/2_ observed as a function of pH. In addition, the delaying effect of ^ox^Aβ_1–40_ on Aβ_1–40_ assembly is kept with similar features as those described in the previous section, regardless of the pH values.

TEM pictures recorded at the end of the two self-assembly processes are shown in Appendix A (experiments N°1 and 3, respectively). They show some differences in the assemblies of Aβ_1–40_ and ^ox^Aβ_1–40_ as a function of pH. We observed the formation of longer and twisted fibrils at a higher pH for Aβ_1–40_ and more fibrillary assemblies at a lower pH for ^ox^Aβ_1–40_, in line with the kinetics of assembly previously described in this work.

The kinetic effect of ^ox^Aβ_1–40_ on Aβ_1–40_ assembly, previously described at pH 7.0, is conserved regardless of the pH values (in the range of 6.0 to 7.4). In addition, as for Aβ_1–40_, the pH has no or little effect on the assembly of the 1:1 stoichiometric mixture of Aβ_1–40_ and ^ox^Aβ_1–40_, followed by t_1/2_ values falling in the same timescale (about 35 h). These results indicate that the strongly pH-dependent rate of ^ox^Aβ_1–40_ assembly has little impact on the co-assembly process at these ratios and concentrations.

### 3.4. Co-Assembly of Aβ_1–40_ and ^ox^Aβ_1–40_ at Various Ratios and Concentrations

Further insights into the co-assembly of Aβ_1–40_ and ^ox^Aβ_1–40_ were provided by incubating the two peptides at various ratios, namely [Aβ_1–40_]/[^ox^Aβ_1–40_] = 20/0, 18/2, 16/4, 10/10, 4/16, 2/18 and 0/20 µM. The results obtained in experiment N°1 at 20/0, 16/4, 10/10, 4/16 and 0/20, along with 4/0, 10/0 and 16/0 for comparison, are shown in Figure 4A–C (see all data gathered in Appendix A for experiments N°1 to 3), and the t_1/2_ and F_max_ values, plotted as a function of the ratio between peptides, are shown in Figure 4D,E. At first glance, these data confirm our previous observations (described for the 1:1 stoichiometric ratio) for all the studied ratios, *id est*, the assembly slowed down with a higher t_1/2_ in the presence of ^ox^Aβ_1–40_, while F_max_ values mainly depended on the Aβ_1–40_ concentration, regardless of the presence of ^ox^Aβ_1–40_. A closer inspection of the t_1/2_, as a function of Aβ_1–40_ concentration, suggests a two-step trend: a significant decrease at low concentrations up to 10 µM, followed by a plateau between 10 and 20 µM. In the presence of ^ox^Aβ_1–40_, the decrease of t_1/2_ with increasing Aβ_1–40_ concentration follows a more linear trend compared to Aβ_1–40_ alone (red versus green dots in Figure 4D). Therefore, the slowdown, due to the presence of ^ox^Aβ_1–40_, is more pronounced at ^ox^Aβ_1–40/_Aβ_1–40_ 1:1 stoichiometric ratio and above. At a lower ^ox^Aβ_1–40/_Aβ_1–40_ ratio, there is still a delaying effect, suggesting that ^ox^Aβ_1–40_ acts on the first nucleation phase. However, the impact of ^ox^Aβ_1–40_ on the slope of s-shape curves is not similarly observed between experiments. Despite an obvious weakening of the slope k, induced by ^ox^Aβ_1–40_ in experiment N°1, this effect is not as clear for experiments N°2 and N°3 (Appendix A). Hence, this will not be commented on here.

Two other sets of experiments were additionally performed, wherein several concentrations of Aβ_1–40_ were added to ^ox^Aβ_1–40_ at 20 µM, and vice versa. The resulting kinetic ThT fluorescence data are shown in Figure 5A (experiment N°1), Appendix A (experiments N°2 and 4) and Appendix A (experiments N°1, 2 and 4), respectively. When Aβ_1–40_ is added to ^ox^Aβ_1–40_, the results are in line with previous observations (Figure 4), as an increased concentration of Aβ_1–40_ is associated with an increase of F_max_ (Figure 5C) and a decrease of t_1/2_ (Figure 5B) values. The concentration dependence of the kinetics of Aβ_1–40_, in the presence of ^ox^Aβ_1–40_ at 20 µM (Figure 5B), differs from that of Aβ_1–40_ (in the absence of ^ox^Aβ_1–40_ at 20 µM, Figure 4D). In addition, a higher positive slope of the co-assembly of Aβ_1–40_ and ^ox^Aβ_1–40_ curves relates to the amount of Aβ_1–40_ added to the samples (Figure 5D). Note that here, k’ has been calculated as the slope between (t(F = F_min_ + ΔF/0.8); F_min_ + ΔF/0.8) and (t(F = F_min_ + ΔF/0.2); F_min_ + ΔF/0.2). The comparison between Aβ_1–40_ in the presence or absence of ^ox^Aβ_1–40_ indicates that ^ox^Aβ_1–40_ may play a dual role; a higher positive slope and increase in t_1/2_, observed in Aβ_1–40_ kinetics, supports an impact on both the nucleation and growth phases, whilst a similar trend is qualitatively observed for experiments N°2 and 4 (Appendix A).

When ^ox^Aβ_1–40_ is added to Aβ_1–40_, the kinetics are weakly affected, mirroring the fact that, above 20 µM in Aβ_1–40_, ^ox^Aβ_1–40_ has little impact on Aβ_1–40_ aggregation. One main difference appears when comparing the results at equimolar ratio (10:10 µM and 20:20 µM, Figure 4B and Figure 5A). Strikingly, adding 10 µM of ^ox^Aβ_1–40_ has a deep slowdown effect on Aβ_1–40_, but not at 20 µM. This indicates that 20 µM (or a concentration in between 10 and 20 µM) represents a threshold value above which ^ox^Aβ_1–40_ is not able to significantly interfere with Aβ_1–40_ assembly.

### 3.5. Impact of Cu(II) and Zn(II) on the Self-Assembly of Aβ_1–40_ and ^ox^Aβ_1–40_ and Their Co-Assembly

-*Self-assembly of Aβ_1–40_:* In the presence of 0.9 equiv. of Cu(II), the Aβ_1–40_ assembly splits into two processes. It starts very rapidly, with a no-lag phase, and reaches a weak fluorescence plateau (approximately 5-fold weaker than the plateau value of apo-Aβ_1–40_ assembly); then, a second sigmoidal process occurs after about 30 h, leading to a final plateau intensity approximately two times lower than the one observed for apo-Aβ_1–40_ (Figure 6 and Appendix A, left, solid and dashed black lines). TEM pictures illustrate the presence of deposits of fibrils that are shorter and thinner than those of the apo-Aβ_1–40_ (Figure 7A, Appendix A). A similar kinetic behavior is observed in the presence of 0.9 equiv. Zn(II) (Figure 6 and Appendix A, right, solid and dashed black lines), but with a much higher fluorescence intensity. Fibrils are mostly detected by TEM at the end of aggregation, in line with the high plateau intensity (Figure 7C, Appendix A). In contrast to the apo-Aβ_1–40_ fibrils, the Zn-Aβ_1–40_ fibrils are untangled and thinner, but form clumps. In presence of either cations, the kinetics of fibrils formation show a quite unusual profile with two distinct phases. We may hypothesize that the first fast-forming aggregates are further reorganized in more stable fibrillar species during the second phase. Note that this two-step trend was observed on all the independent experiments.-*Self-assembly of ^ox^Aβ_1–40_:* In the presence of 0.9 equiv. of Cu(II), the ThT fluorescence curve of ^ox^Aβ_1–40_ shows a very weak intensity that appears after a rapid increase (Figure 6, left, blue solid line and inset). No sigmoidal process is observed at the time scale of the experiment in contrast to Cu-Aβ_1–40_ aggregation (Figure 6, black line). Although the final plateau of ThT fluorescence intensity is very low, the TEM pictures show the presence of oligomeric species and amorphous aggregates, along with some fibrils of various morphologies (Figure 7B, Appendix A). In line with the observation made for the apo peptides, their formation may be triggered by the presence of the remaining Aβ_1–40_ in the sample. The apparent divergence between ThT fluorescence and TEM results may also originate from the formation of fibrils (i) having weak interaction with ThT, (ii) having interaction with ThT but giving rise to a low fluorescence enhancement or (iii) with the quenching of the ThT fluorescence by the Cu(II) paramagnetism, since the Cu(II) site is altered by the oxidation of the peptide [70]. In the presence of 0.9 equiv. Zn(II), the ThT fluorescence intensity rapidly reaches its maximal value, which is quite weak compared to Zn-Aβ_1–40_ (about 5-times lower). In line with the lower fluorescence plateau value, less fibrils are observed by TEM, where dense deposits of amorphous aggregates are also present (Figure 7D, Appendix A).-*Co-assembly:* Furthermore, we also studied the assembly behavior of an equimolar mixture of Aβ_1–40_ and ^ox^Aβ_1–40_ in the presence of 0.9 equiv. of Cu(II) or Zn(II) (Figure 6, solid green lines). With Cu(II), the two-step kinetics is recovered, with a first plateau value weaker than the one for Cu-Aβ_1–40_, and a second sigmoidal process that takes more time to occur (about 60 h versus 20 h). This is reminiscent of what was previously reported in the case of the self-assembly of Cu_2_-Aβ_1–40_ (in the presence of 2 Cu(II) ion per Aβ_1–40_ peptide), and suggests that Cu(II) is weakly bound to ^ox^Aβ_1–40_ (at least weaker than the second site in Aβ_1–40_) and that it can mainly be transferred to Aβ_1–40_ [21]. The formation of ternary ^ox^Aβ_1–40_-Cu-Aβ_1–40_ is also possible. With Zn(II), the kinetic trace is half-way from that of Zn-Aβ_1–40_ and Zn-^ox^Aβ_1–40_, in line with various possible events at play (independent assembly of Zn-Aβ_1–40_ and Zn-^ox^Aβ_1–40_, or the assembly of Zn_2_-Aβ_1–40_), prevents deeper analysis.

**Figure 6 antioxidants-12-00472-f006:**
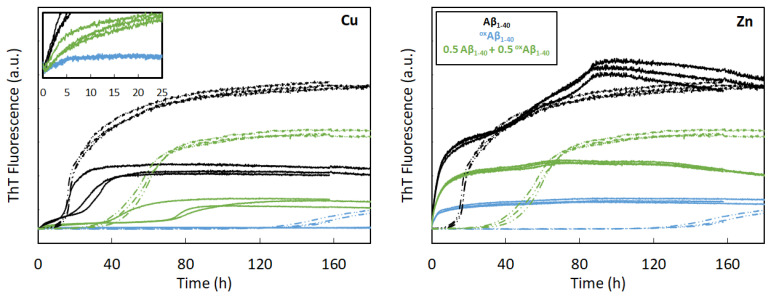
Kinetic monitoring of Aβ_1–40_ and ^ox^Aβ_1–40_ co-assembly in presence of Cu(II) or Zn(II) using ThT fluorescence. Curves are shown as triplicate. [Aβ_1–40_] = 20 µM (black); [Aβ_1–40_] = [^ox^Aβ_1–40_] = 10 µM (green); [ ^ox^Aβ_1–40_] = 20 µM (blue). Apo peptides (dashed curves) with Cu(II) (18 µM, solid curves, left panel) or Zn(II) (18 µM, solid curves, right panel) at pH 7.4. Data from Experiment N°1: HEPES buffer 50 mM, NaCl 65 mM. Y-axis corresponds to ThT fluorescence in arbitrary unit (a.u.); data are directly comparable between them (same y-scale).

**Figure 7 antioxidants-12-00472-f007:**
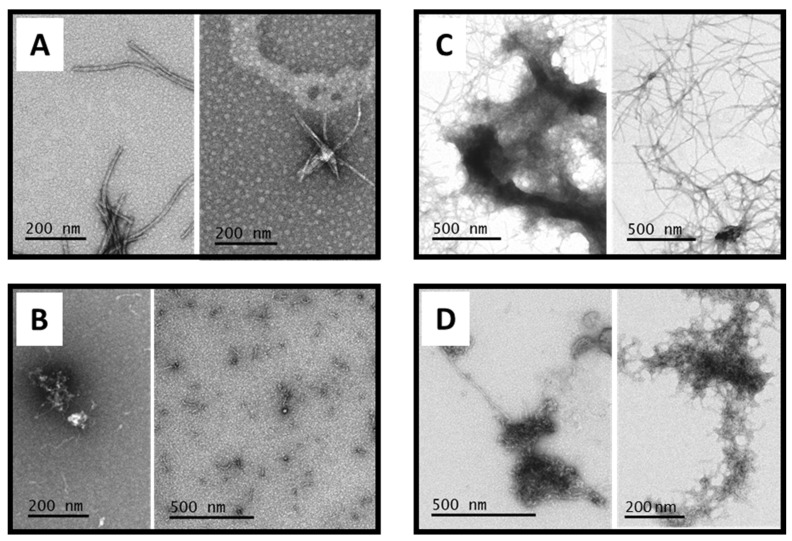
Selected TEM pictures of Cu(Aβ_1–40_) and Cu(^ox^Aβ_1–40_) at 20 µM ((**A**) and (**B**), respectively) and of Zn(Aβ_1–40_) and Zn(^ox^Aβ_1–40_) at 20 µM ((**C**) and (**D**), respectively) taken at the end of the ThT fluorescence experiment N°1. Hepes buffer 50 mM, [NaCl] = 65 mM, pH 7.4. Two shots are given to better illustrate the heterogeneity of the assemblies formed.

## 4. Concluding Remarks

### 4.1. Reproducibility Issues

We acknowledge that batch effects are the first cause of non-reproducible data in amyloid kinetics assessments in vitro [12,79]. However, the converging results from our various datasets emphasize that the assembly trends and their response to multiple stimuli (pH, metal ions, ratio of peptides in their mixture …) discussed in this work are consistent. Furthermore, the trends observed in the presence of ^ox^Aβ_1–40_ are known to be dependent on the level of oxidation that can fluctuate between experiments. Hence, it is quite remarkable that, despite these two sources of variability, the trends we observed in both the kinetics and amyloids morphology point toward the same conclusions.

### 4.2. Ability of ^ox^Aβ_1–40_ to Self-Assemble and Importance of the Oxidation Paths of Aβ_1–40_

The present study shows that the oxidative damages undergone by Aβ_1–40_ during the ROS production, catalyzed by the Cu-Aβ_1–40_, in the presence of ascorbate as reductant and dioxygen, have a strong impact on the assembly processes. To our knowledge, there is no report on ^ox^Aβ_1–40_ with multiple oxidation sites mirroring the biological complexity, including oxidation on the His to form the 2-oxo-His. Indeed, in contrast to the previous studies, ^ox^Aβ_1–40_ refers to a set of oxidized Aβ_1–40_ species, which includes oxidative modifications mainly on Asp1, His13 and His14 in our experimental conditions [35], and which are more biologically relevant than the site-specific oxidative modifications studied before [36,37,38,39,40,41,42,43,44]. This difference in the oxidation process participates in some divergent data, with respect to the literature [63,64,65,66,67,68,69], whilst the previous trend of a slow-down of the assembly, induced by oxidation, is also obtained here. The ^ox^Aβ_1–40_ species keep their ability to aggregate, but at a much lower rate, and generate amorphous and smaller assemblies rather than long and thick β-sheet rich fibrils, usually observed with non-oxidized Aβ_1–40_. In a very recent paper [20], Tian and Viles found that the pKa values (about 6.7) of the three His residues [82,83] of Aβ_1–40_, rather than the pI (5.3), play a substantial role in the modification of the kinetic assembly trends, with disruption of the electrostatic interactions due to protonated His residues, leading to weaker primary nucleation. In the context of ^ox^Aβ_1–40_, 2-oxo-His are neutral and are formed from pH well beyond 6.7 [84]. Hence, this may explain the lower self-assembly propensity of ^ox^Aβ_1–40_ compared to Aβ_1–40_. In the presence of Cu(II) or Zn(II) ion, the extent of ^ox^Aβ_1–40_ assembly is also much weaker than that of Aβ_1–40_.

### 4.3. Effect of ^ox^Aβ_1–40_ on Aβ_1–40_ Assembly

The oxidation of the peptide has a strong impact on its assembly (Figure 3), as shown in both our kinetic and morphology assessments. We demonstrated the fact that, despite a range of 10–40% of non-oxidized Aβ_1–40_ peptide remaining in ^ox^Aβ_1–40_ samples, ^ox^Aβ_1–40_ was able to delay the Aβ_1–40_ assembly that is not driven by a dilution effect. Given this, we demonstrated a modulatory activity of the ^ox^Aβ_1–40_ peptides towards the assembly of Aβ_1–40_: a delaying effect with a reduction of the t_1/2_ and an increase in the growth rate. Interestingly, this delaying effect is partially observed in the presence of Cu(II) (on the second sigmoidal step), but not in the presence of Zn(II), suggesting that the reorganization towards more fibrillary architectures is hampered by the formation of the amorphous ^ox^Aβ_1–40_-Zn species.

### 4.4. Effect of pH

We found that ^ox^Aβ_1–40_ delays the Aβ_1–40_ assembly in a wide range of pH (6.0–7.4), while a strong pH dependence of ^ox^Aβ_1–40_ assembly is observed, id est, there is a much slower self-assembly at pH 7.4 compared to pH 6.0, induced by deprotonation of remaining non-oxidized His residues or N-terminal amine. This may be explained by the disruption of electrostatic interactions [21], as reported for related amyloid-forming peptides [20] and for peptides approaching global neutrality with decreasing pH [85].

### 4.5. Mechanistic Insights

Here, we suggest a tentative mechanism, taking into account our findings, illustrated in Figure 3. On the left side, the self-assembly of Aβ_1–40_ and ^ox^Aβ_1–40_ are compared. Upon oxidation, the peptide loses most of its ability to form ThT-responsive fibrils (lower level of β-sheet rich fibrils, in blue) and their formation takes a longer time (clock), while the formation of ill-defined intermediate size species is observed (higher level of off-pathways assemblies, in grey). On the right side, the co-assembly of Aβ_1–40_ and ^ox^Aβ_1–40_ (both at 10 µM) is compared to the self-assembly of Aβ_1–40_ at 10 µM. The same levels of fibrils are formed, but their formation takes longer in the presence of ^ox^Aβ_1–40_ (clock), while the level of off-pathways is higher.

Further mechanistic investigations to gain more insights into the role of Cu(II) and Zn(II) include a pH-dependent study of the effect of the metal ions on the assembly and co-assembly of Aβ_1–40_ and ^ox^Aβ_1–40_, which is currently under progress in our lab.

### 4.6. Biological Relevance

The results shown here are in line with recent reports showing the impact of Aβ oxidation on aggregation [43,86], although the authors mainly focused on the effect of Met35 oxidation [65,67,87] and emphasized the intricacy of the connection between the different aggregation effectors studied here (Aβ oxidation and metal ions). One general trend is the observation of more heterogeneous and smaller size aggregates with the ^ox^Aβ_1–40_ peptide, leading to weaker ThT fluorescence, including in the presence of metal ions. This might have fallouts with respect to the toxicity of Aβ-based aggregates, since it is now quite well-accepted that oligomers are more toxic than mature fibrils [47,88].

With respect to the biological relevance, further work could include the study of the impact of Fe(II) on the assembly and co-assembly of Aβ_1–40_ and ^ox^Aβ_1–40_ to complement Cu(II) and Zn(II) studied here.

## Data Availability

Data is contained within the article and Appendix A.

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
