# Peer review of "Oxidative Damages on the Alzheimer’s Related-Aβ Peptide Alters Its Ability to Assemble"

_antioxidants, 2023, doi:10.3390/antiox12020472_

Round 1

Reviewer 1 Report

Oxidative stress results in the oxidation of amyloid-β (Aβ) peptides. In this study, the author studies the influence of oxidation in Aβ assembly. Cu(II) or Zn(II) are in vitro modulators of Aβ assembly. 

Major comment:

1. Is there a difference between PH and Aβ/oxAβ in the presence of Cu(II) or Zn(II)?

2. is the other cation, Fe(II), have any influence in the Aβ/oxAβ ? 

Minor comment: It will be easier for reading to indicate the Aβ or oxAβ in the line of Figure 6.

Author Response

(x) Extensive editing of English language and style required

ANSWER: The manuscript has been fully revised by a native English speaker working in the field of peptide self-assembly and associated disorders, who is now acknowledged in the revised version.

Major comment:

  1. Is there a difference between PH and Aβ/oxAβ in the presence of Cu(II) or Zn(II)?

ANSWER: This is an interesting question. We have decided to keep close to the physiological pH values, for a matter of biological relevance. Such mechanistically-based study could clearly be the topic of a forthcoming study and manuscript. But we want to point out that this is beyond the scope of the current article (for a matter of comparison, a recent paper was published in Angew. Chem. Int. Ed. that focused only on the impact of pH of Aβ self-assembly). We have added a sentence in the concluding section about such highly interesting perspectives of the current work. “Further mechanistic investigations to gain more insights into the role of Cu(II) and Zn(II) include a pH-dependent study of the effect of the metal ions on the assembly and co-assembly of Aβ1-40 and oxAβ1-40, which is currently under progress is our lab.” 

  1. is the other cation, Fe(II), have any influence in the Aβ/oxAβ

ANSWER: This is another highly relevant question. We did not address this point because we still have not studied the impact of Fe on Ab oxidation. In addition, we consider that iron is not the main relevant ion, so we focused on Copper and Zinc. This is mainly due to the fact that (i) iron is found bound as nanoparticles in senile plaques and (ii) there is very few report on the possibility to have a Aβ – Fe(II) molecular interaction, (iii) To the best of our knowledge, there is no report on the impact of Fe(II) on the Aβ self-assembly. This is likely because this requires, as for Cu(I), a dedicated setup to keep the experiment under inert atmosphere. Else Fe(II) is rapidly oxidized to Fe(III) making strongly stable Fe(OH)3 that precipitate. This point is now addressed in the introduction with relevant references quoted and also introduced as a possible forthcoming work in the concluding section. “Additionally, iron impairment is another contributor to AD pathology which was found in senile plaques.[49,50] However, its exact role is not fully understood[50,51] and its speciation is not clear, with mainly nanoparticles of Fe such as magnetite or ferritine-based mineral being detected in the core of the plaques.[49] In contrast to Cu and Zn for which the molecular interaction with Aβ was characterized, only few studies propose a possible coordination site for the Fe(II)-Aβ,[52] or Fe(III)-Aβ complex.[53] Despite the Fe(II)/Fe(III) redox couple may participate to the oxidative stress linked to AD,[54,55] there is no evidence that this may be due to the Fe(II)/Fe(III)-Aβ interaction. There is also no study on the Fe(II) impact on the Aβ self-assembly, even if rare findings in this context were obtained for Fe(III)[56-59] or ferritin nanoparticles.[60] Therefore, the present report, is focused on the impact of the metal ions Cu(II) and Zn(II) on the assembly of Aβ.” & “With respect to the biological relevance, further work could include the study of the impact of Fe(II) on the assembly and co-assembly of Aβ1-40 and oxAβ1-40 to complement to Cu(II) and Zn(II) studied here.”        

Minor comment: It will be easier for reading to indicate the Aβ or oxAβ in the line of Figure 6.

This has been corrected.

Reviewer 2 Report

The manuscript by Cheignon et al. describes the research on the influence of Aβ1-40 peptide oxidation on its aggregation. The experiments are well-planned and quite thoroughly analyzed. The results are probably relevant for Alzheimer's disease development. However, there are several issues to be clarified:

1. There is no result shown in the manuscript proving the oxidation of the Aβ1-40 peptide used by the Authors. Obtained HPLC chromatograms and HR-MS spectra should be presented, at least in the Supplementary Materials. The spectra and chromatograms should be supplemented with the list of the most intense ions detected by MS, with the appropriate assignment to the number of oxidized residues in the peptide.

2. Lines 21-22: The Authors wrote: "In addition, oxAβ was found to slow down the formation of Aβ fibril without having a strong impact on the quantity and morphology of the Aβ fibrils formed." The reviewer recommends clarifying that this influence is on non-oxidized Aβ fibril formation.

3. Lines 258 and 297: The Authors use the term "stoichiometric ratio" instead of "molar ratio." The stoichiometric ratio is not always 1:1, but this is what was probably meant in these fragments. Please clarify and write unequivocally that it is a 1:1 (or other) molar ratio.

4. Figure 5: The traces for 20 μM Aβ1-40 alone (black) are not shown, contrary to what is written in the caption.

5. Lines 321, 334, and 344: Figure S8-S9 should be Figure S9-S10.

6. Line 346: S7 should be S8.

7. Figure S8: this figure does not show traces for co-assembly but shows only assembly of non-oxidized alone (plus Cu or Zn) or oxidized (plus Cu or Zn) peptides. There are no traces of mixtures of non-oxidized with oxidized peptides (contrary to the caption and the main text).

8. Line 525: "between 150 and 2000 Da" should be "between 150 and 2000 m/z."

Author Response

#2

  1. There is no result shown in the manuscript proving the oxidation of the Aβ1-40 peptide used by the Authors. Obtained HPLC chromatograms and HR-MS spectra should be presented, at least in the Supplementary Materials. The spectra and chromatograms should be supplemented with the list of the most intense ions detected by MS, with the appropriate assignment to the number of oxidized residues in the peptide.

ANSWER: A representative chromatogram is given as a matter of the illustration as Figure S0 and is introduced in the text “According to ref.[35], Chromatographic peaks were integrated, and the remaining Aβ1-40 in oxAβ1-40 samples was found to be in the 10-40 % range depending on the performed experiment, as illustrated in Figure S0 for experiment 1 for which 15% ± 3 of remaining Aβ1-40 was evaluated.”

The full analysis of peptide oxidation has been previously reported in ref 35, which is quoted at relevant places.

  1. Lines 21-22: The Authors wrote: "In addition, oxAβ was found to slow down the formation of Aβ fibril without having a strong impact on the quantity and morphology of the Aβ fibrils formed." The reviewer recommends clarifying that this influence is on non-oxidized Aβ fibril formation.

ANSWER: This has been clarified as “In addition, oxAβ does affect the assembly of the parent Ab peptide. In mixture of the two peptides, oxAβ has mainly a kinetic effect on the assembly of the Ab peptide: oxAβ slows down the formation of Ab fibril but doesn’t change the quantity and morphology of the Aβ fibrils formed to a significant extent.”    

  1. Lines 258 and 297: The Authors use the term "stoichiometric ratio" instead of "molar ratio." The stoichiometric ratio is not always 1:1, but this is what was probably meant in these fragments. Please clarify and write unequivocally that it is a 1:1 (or other) molar ratio.

ANBSWER: We thank the reviewer for this clarification. This was checked and changed accordingly accordingly. 

  1. Figure 5: The traces for 20 μM Aβ1-40 alone (black) are not shown, contrary to what is written in the caption.

ANSWER: This has been corrected.

  1. Lines 321, 334, and 344: Figure S8-S9 should be Figure S9-S10.

ANSWER: This has been changed.

  1. Line 346: S7 should be S8.

ANSWER: This has been changed.

  1. Figure S8: this figure does not show traces for co-assembly but shows only assembly of non-oxidized alone (plus Cu or Zn) or oxidized (plus Cu or Zn) peptides. There are no traces of mixtures of non-oxidized with oxidized peptides (contrary to the caption and the main text).

ANSWER: This has been corrected.

  1. Line 525: "between 150 and 2000 Da" should be "between 150 and 2000 m/z."

ANSWER: This has been corrected.

Round 2

Reviewer 2 Report

Unfortunately, the Authors did not respond appropriately to the first and the most critical objection of the reviewer, which was as follows:

"

1. There is no result shown in the manuscript proving the oxidation of the Aβ1-40 peptide used by the Authors. Obtained HPLC chromatograms and HR-MS spectra should be presented, at least in the Supplementary Materials. The spectra and chromatograms should be supplemented with the list of the most intense ions detected by MS, with the appropriate assignment to the number of oxidized residues in the peptide.

"

In response to that point, the Authors showed only the proof of the disappearance of the Aβ1-40 peptide. It is unacceptable and significantly lowers the usefulness of the manuscript. The whole manuscript is built on the assumption that the peptide is oxidized, but the Authors did not show any proof of this modification in their results. Oxidative conditions may lead to many peptide modifications, such as incorporating oxygen atoms into the side chain, but they may also lead to peptide cleavage. Each of these modifications leads to different conclusions. Thus, the Authors must show results documenting the nature of the modification, which residues were modified, and how many (average) modifications per peptide molecule were introduced in each experiment (see the point from the previous review report). Otherwise, the manuscript is too speculative, and the reviewer does not recommend publishing it in Antioxidants.

Author Response

Unfortunately, the Authors did not respond appropriately to the first and the most critical objection of the reviewer, which was as follows:

"

  1. There is no result shown in the manuscript proving the oxidation of the Aβ1-40 peptide used by the Authors. Obtained HPLC chromatograms and HR-MS spectra should be presented, at least in the Supplementary Materials. The spectra and chromatograms should be supplemented with the list of the most intense ions detected by MS, with the appropriate assignment to the number of oxidized residues in the peptide.

"

In response to that point, the Authors showed only the proof of the disappearance of the Aβ1-40 peptide. It is unacceptable and significantly lowers the usefulness of the manuscript. The whole manuscript is built on the assumption that the peptide is oxidized, but the Authors did not show any proof of this modification in their results. Oxidative conditions may lead to many peptide modifications, such as incorporating oxygen atoms into the side chain, but they may also lead to peptide cleavage. Each of these modifications leads to different conclusions. Thus, the Authors must show results documenting the nature of the modification, which residues were modified, and how many (average) modifications per peptide molecule were introduced in each experiment (see the point from the previous review report). Otherwise, the manuscript is too speculative, and the reviewer does not recommend publishing it in Antioxidants.

ANSWER: To fully answer the revision asked by the reviewer, we have now included the requested data as Figures S1-S2, which corresponds to the oxidative damages of the Aβ1-40 peptide after 30 minutes, that is to say when the oxidation is stopped and the oxAβ1-40 recovered. The data are introduced line 159-64 (yellow in the text) and in the description of the supplementary materials.

” Oxidative damages leading to oxAβ1-40.

                The detailed and full kinetic analysis of the oxidative damages underwent by the Aβ1-40 peptide has been reported previously.[35] Hence the characterizations of the oxidation of the Aβ1-40 peptide is given in the Supporting Information (Figures S1-S2). The results are fully consistent with those reported previously on another peptide batch,[35] in line with the correct replicability of such Cu-Aβ/ascorbate/dioxygen oxidation procedure.[34]”

“Supplementary Materials: quantification of remaining Aβ1-40 after oxidation of Aβ1-40 ; mass spectrometry analysis of Aβ1-40 oxidation ; complementary Aβ1-40 and ox1-40 assembly curves and TEM pictures as function of pH, of mixture of Aβ1-40 and ox1-40,and in presence of Cu(II) and Zn(II) ions.”

In addition, we have previously and fully characterized the oxidation of the Aβ1-40 peptide (Cheignon et al., Inorganica Chimica Acta, 2018, ref 35 in the current manuscript) as well as the shorter peptide Aβ1-28 (Cassagnes et al, Angew. Chem. Int. Ed, 2013, ref 34 in the current manuscript). The reproducibility of the oxidative damages is robust and the oxidative damages obtained with the Asc/Cu/Aβ1-40 are shown in Figure 1 together with reported oxidative damages described in other groups under different oxidation conditions. This point has been clarified in the manuscript (in yellow in the text).

Lines 130-5: “In addition to those studies where oxidation at one specific amino-acid residues was explored, our main focus was to study (i) whether and to what extent the biologically relevant combination of oxidized peptides (noted ox1-40)[31,34,35,70,71] formed by the Cu-Aβ / ascorbate / dioxygen triad [35] self-assembles and modulates the assembly of Aβ1-40 and (ii) the difference in the metal-modulated assembly of ox1-40 compared to Aβ1-40 peptides.”

Line 142 : …. whilst the detailed analysis of ox1-40 formation was previously reported.[35]